# National and regional seasonal dynamics of all-cause and cause-specific mortality in the USA from 1980 to 2016

Robbie M Parks[1,2], James E Bennett[1,2,3], Kyle J Foreman[1,2,4], Ralf Toumi[5], Majid Ezzati[1,2,3]*

[1]MRC-PHE Centre for Environment and Health, Imperial College London, London, United Kingdom; [2]Department of Epidemiology and Biostatistics, School of Public Health, Imperial College London, London, United Kingdom; [3]WHO Collaborating Centre on NCD Surveillance and Epidemiology, Imperial College London, London, United Kingdom; [4]Institute for Health Metrics and Evaluation, University of Washington, Seattle, United States; [5]Space and Atmospheric Physics, Imperial College London, London, United Kingdom

**Abstract** In temperate climates, winter deaths exceed summer ones. However, there is limited information on the timing and the relative magnitudes of maximum and minimum mortality, by local climate, age group, sex and medical cause of death. We used geo-coded mortality data and wavelets to analyse the seasonality of mortality by age group and sex from 1980 to 2016 in the USA and its subnational climatic regions. Death rates in men and women $\geq$ 45 years peaked in December to February and were lowest in June to August, driven by cardiorespiratory diseases and injuries. In these ages, percent difference in death rates between peak and minimum months did not vary across climate regions, nor changed from 1980 to 2016. Under five years, seasonality of all-cause mortality largely disappeared after the 1990s. In adolescents and young adults, especially in males, death rates peaked in June/July and were lowest in December/January, driven by injury deaths.

DOI: https://doi.org/10.7554/eLife.35500.001

*For correspondence:
majid.ezzati@imperial.ac.uk

## Introduction

It is well-established that death rates vary throughout the year, and in temperate climates there tend to be more deaths in winter than in summer (*Campbell, 2017*; *Fowler et al., 2015*; *Healy, 2003*; *McKee, 1989*). It has therefore been hypothesized that a warmer world may lower winter mortality in temperate climates (*Langford and Bentham, 1995*; *Martens, 1998*). In a large country like the USA, which possesses distinct climate regions, the seasonality of mortality may vary geographically, due to geographical variations in mortality, localized weather patterns, and regional differences in adaptation measures such as heating, air conditioning and healthcare (*Davis et al., 2004*; *Braga et al., 2001*; *Kalkstein, 2013*; *Medina-Ramón and Schwartz, 2007*). The presence and extent of seasonal variation in mortality may also itself change over time (*Bobb et al., 2014*; *Carson et al., 2006*; *Seretakis et al., 1997*; *Sheridan et al., 2009*).

A thorough understanding of the long-term dynamics of seasonality of mortality, and its geographical and demographic patterns, is needed to identify at-risk groups, plan responses at the present time as well as under changing climate conditions. Although mortality seasonality is well-established, there is limited information on how seasonality, including the timing of minimum and maximum mortality, varies by local climate and how these features have changed over time, especially in relation to age group, sex and medical cause of death (*Rau, 2004*; *Rau et al., 2018*).

**eLife digest** In the USA, more deaths happen in the winter than the summer. But when deaths occur varies greatly by sex, age, cause of death, and possibly region. Seasonal differences in death rates can change over time due to changes in factors that cause disease or affect treatment.

Analyzing the seasonality of deaths can help scientists determine whether interventions to minimize deaths during a certain time of year are needed, or whether existing ones are effective. Scrutinizing seasonal patterns in death over time can also help scientists determine whether large-scale weather or climate changes are affecting the seasonality of death.

Now, Parks et al. show that there are age and sex differences in which times of year most deaths occur. Parks et al. analyzed data on US deaths between 1980 and 2016. While overall deaths in a year were highest in winter and lowest in summer, a greater number of young men died during summer – mainly due to injuries – than during winter. Seasonal differences in deaths among young children have largely disappeared and seasonal differences in the deaths of older children and young adults have become smaller. Deaths among women and men aged 45 or older peaked between December and February – largely caused by respiratory and heart diseases, or injuries. Deaths in this older age group were lowest during the summer months. Death patterns in older people changed little over time. No regional differences were found in seasonal death patterns, despite large climate variation across the USA.

The analysis by Parks et al. suggests public health and medical interventions have been successful in reducing seasonal deaths among many groups. But more needs to be done to address seasonal differences in deaths among older adults. For example, by boosting flu vaccination rates, providing warnings about severe weather and better insulation for homes. Using technology like hands-free communication devices or home visits to help keep vulnerable elderly people connected during the winter months may also help.

DOI: https://doi.org/10.7554/eLife.35500.002

In this paper, we comprehensively characterize the spatial and temporal patterns of all-cause and cause-specific mortality seasonality in the USA by sex and age group, through the application of wavelet analytical techniques, to over three decades of national mortality data. Wavelets have been used to study the dynamics of weather phenomena (*Moy et al., 2002*) and infectious diseases (*Grenfell et al., 2001*). We also used centre of gravity analysis and circular statistics methods to understand the timing of maximum and minimum mortality. In addition, we identify how the percentage difference between death rates in maximum and minimum mortality months has changed over time.

## Results

*Table 1* presents number of deaths by cause of death and sex. Deaths from cardiorespiratory diseases make up nearly half of all deaths (48.1%), with most deaths in that group from cardiovascular diseases. Next highest during the study period were deaths from cancers (23.2%), followed by injuries (6.8%), with two thirds of those being from unintentional injuries.

All-cause mortality in males had a 12 month seasonality in all age groups, except ages 35–44 years, for whom there was periodicity at 6 months (*Figure 1*). In females, there was 12 month seasonality in all groups except 5–14 and 25–34 years (p-values=0.21 and 0.25, respectively) (*Figure 2*). While seasonality persisted throughout the entire analysis period in older ages, it largely disappeared after the late 1990s in children aged 0–4 years in both sexes and in women aged 15–24 years.

Deaths from all causes of death were seasonal in older adults (above 65 or 75 years depending on cause, p-values<0.06) (*Figures 1–10* and respective figure supplements), except for intentional injuries and substance use disorders. Deaths from cardiorespiratory diseases, and within it respiratory infections, exhibited seasonality throughout the life-course (p-values<0.06) except for males aged 5–24 years and females aged 15–24 years (p-values>0.11). In addition to older ages, injury deaths were seasonal from childhood through 44 years in women and through 64 years in men (p-values<0.09). Unintentional injuries drove the seasonality of injury deaths for females, whereas both

**Table 1.** Number of deaths, by cause of death and sex from 1980 to 2016.

| Cause | | | Male | Female | Total |
|---|---|---|---|---|---|
| All cause | | | 43,558,203 | 42,295,973 | **85,854,176** |
| | Cancers | | 10,481,582 | 9,476,530 | 19,958,112 |
| | Cardiorespiratory diseases | | 20,168,049 | 21,109,525 | 41,277,574 |
| | | Cardiovascular diseases | 16,238,344 | 17,210,556 | 33,448,900 |
| | | Chronic respiratory diseases | 2,791,652 | 2,595,950 | 5,387,602 |
| | | Respiratory infections | 1,138,053 | 1,303,019 | 2,441,072 |
| | Injuries | | 4,034,876 | 1,768,170 | 5,803,046 |
| | | Unintentional | 2,489,142 | 1,348,187 | 3,837,329 |
| | | Intentional | 1,545,734 | 419,983 | 1,965,717 |
| | Other causes | | 8,873,696 | 9,941,748 | 18,815,444 |

DOI: https://doi.org/10.7554/eLife.35500.003

unintentional and intentional injuries were seasonal in males in most ages, with the exception of below 15 years and above 85 years when intentional injuries were not seasonal (*Figure 7—figure supplement 1*). Consistent seasonality in cancer deaths (*Figures 3,4*) only appeared after 55 years of age (p-values<0.05). No consistent seasonality was evident in substance use disorders (*Figure 9—figure supplement 1* and *Figure 10—figure supplement 1*) or maternal conditions (*Figure 10—figure supplement 2*).

Centre of gravity analysis showed that death rates in men aged ≥45 years and women aged ≥35 years peaked in December, January or February and were lowest in June to August, for all-cause mortality as well as for all non-injury and non-maternal causes of death (*Figure 11* and respective figure supplements). Deaths from cardiorespiratory diseases, including cardiovascular diseases, chronic respiratory diseases and respiratory infections, were also consistently highest in January and February and lowest in July and August across all ages, except for chronic respiratory diseases in ages 5–24 years where there are few deaths from this cause leading to unstable estimates (p-values for seasonality from wavelet analysis ranged from 0.35 to 0.48 for these ages). A similar temporal pattern was seen for all-cause and non-injury mortality in children younger than five years of age, whose all-cause death rate was highest in February and lowest in August. In contrast, among males aged 5–34 years, all-cause mortality peaked in June or July, as did deaths from injuries, which generally had a summer peak in males and females below 45 years of age.

From 1980 to 2016, the proportional (percent) difference in all-cause death rates between peak and minimum months declined little for people older than 45 years of age (by less than eight percentage points with p-values for declining trend >0.1) (*Figure 12*). In contrast, the difference between peak (summer) and minimum (winter) death rates declined in younger ages, by over 25 percentage points in males aged 5–14 years and 15–24 years (p-values<0.01), largely driven in the declining difference between summer and winter injury deaths. Under five years of age, percent seasonal difference in all-cause death rates declined by 13 percentage points (p-value<0.01) for boys but only five percentage points (p-value=0.12) for girls. These declines in seasonality of child deaths were a net effect of declining winter-summer difference in cardiorespiratory diseases deaths and increasing summer-winter difference in injury deaths, itself driven by increasing difference in non-intentional injuries (*Figure 12—figure supplement 1*). Within specific cardiorespiratory diseases in under-five children, percent difference declined for cardiorespiratory diseases, cardiovascular diseases, and chronic respiratory diseases while increasing for respiratory infections.

The subnational centre of gravity analysis showed that all-cause mortality peaks and minima in different climate regions are consistent with the national ones (*Figures 13–16*), indicating that seasonality is largely independent of geography. The relative homogeneity of the timing of maximum and minimum mortality contrasts with the large variation in seasonal temperatures among climate regions. For example, in men and women aged 65–74 years, all-cause mortality peaked in February in the Northeast and Southeast, even though the average temperatures for those regions were different by over 13 degrees Celsius (9.3 in the Southeast compared with −3.8 in the Northeast).

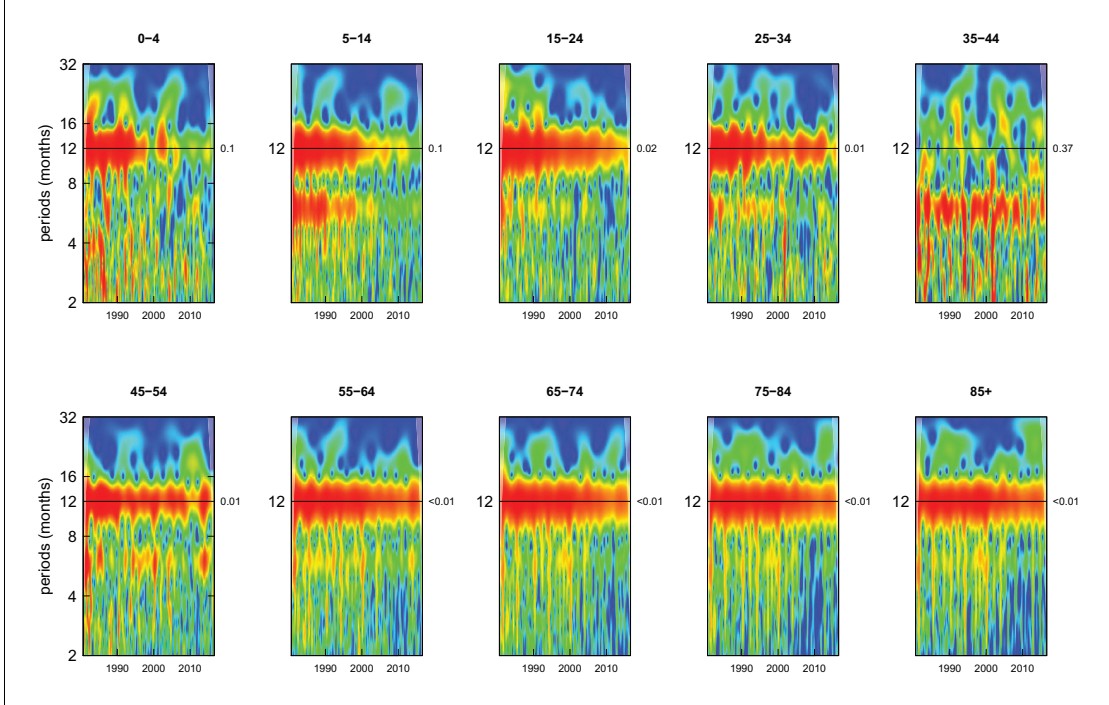

**Figure 1.** Wavelet power spectra for national time series of all-cause death rates for 1980–2016, by age group for males. Wavelet power values increase from blue to red. The shaded regions at the left and right edge of each box indicate the cone of influence, where spectral analysis is less robust. P-values for the presence of 12 month seasonality are to the right of each figure at the 12 month line.

DOI: https://doi.org/10.7554/eLife.35500.004

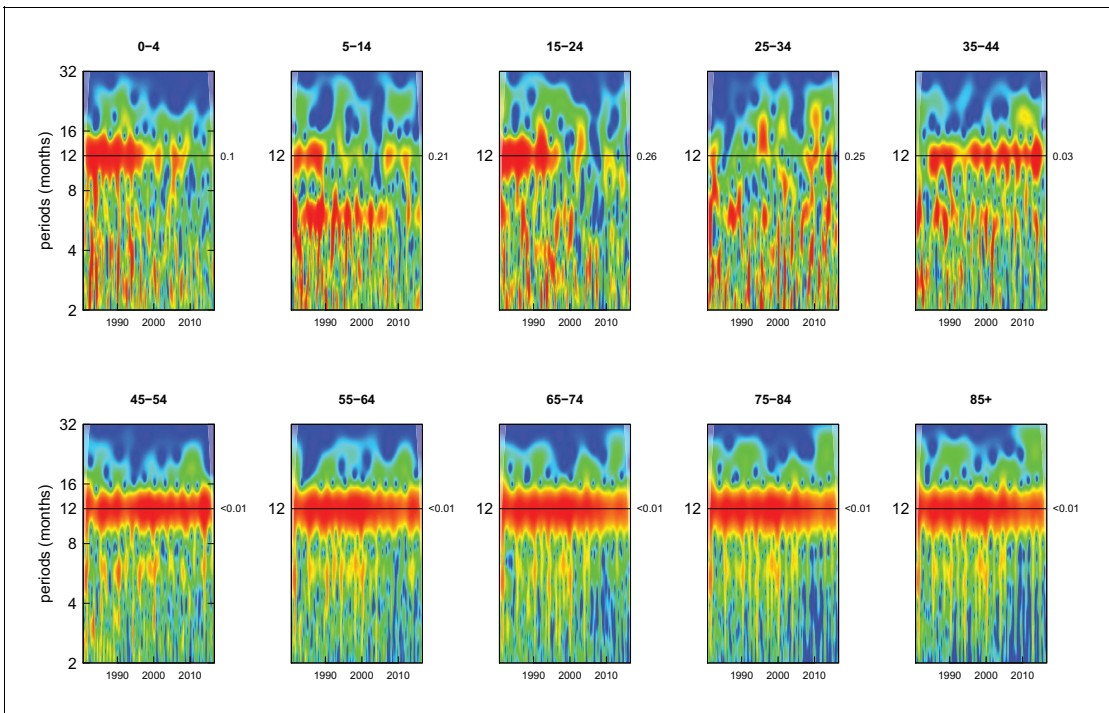

**Figure 2.** Wavelet power spectra for national time series of all-cause death rates for 1980–2016, by age group for females. Wavelet power values increase from blue to red. The shaded regions at the left and right edge of each box indicate the cone of influence, where spectral analysis is less robust. P-values for the presence of 12 month seasonality are to the right of each figure at the 12 month line.

DOI: https://doi.org/10.7554/eLife.35500.005

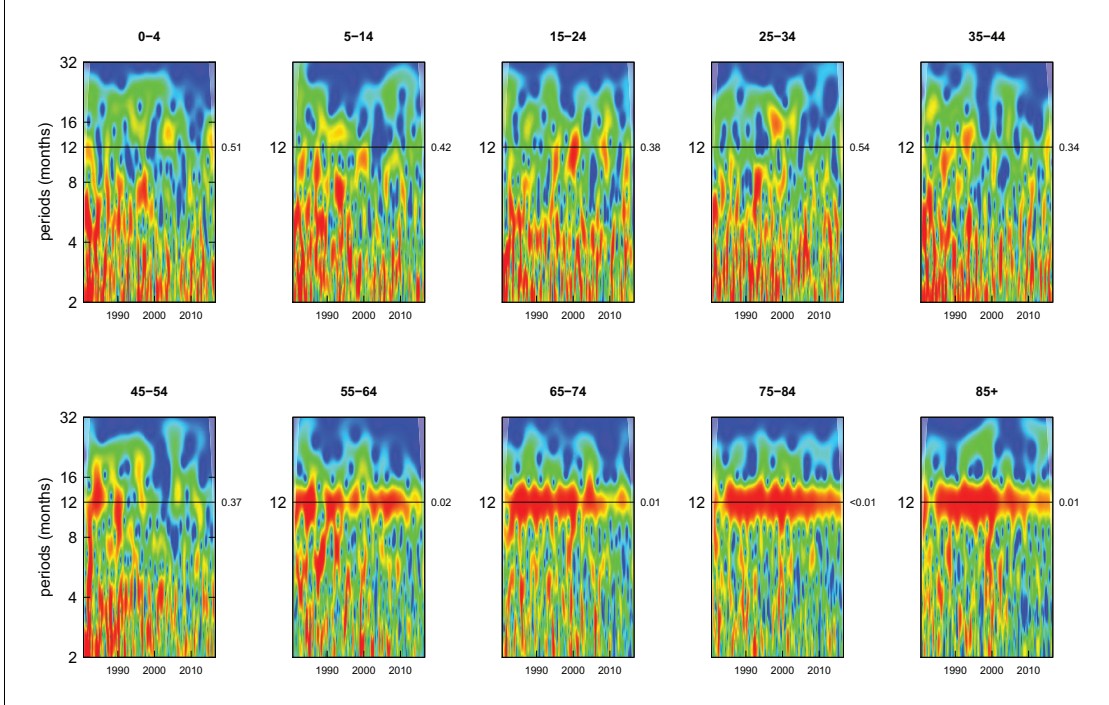

**Figure 3.** Wavelet power spectra for national time series of cancer death rates for 1980–2016, by age group for males. Wavelet power values increase from blue to red. The shaded regions at the left and right edge of each box indicate the cone of influence, where spectral analysis is less robust. P-values for the presence of 12 month seasonality are to the right of each figure at the 12 month line.

DOI: https://doi.org/10.7554/eLife.35500.006

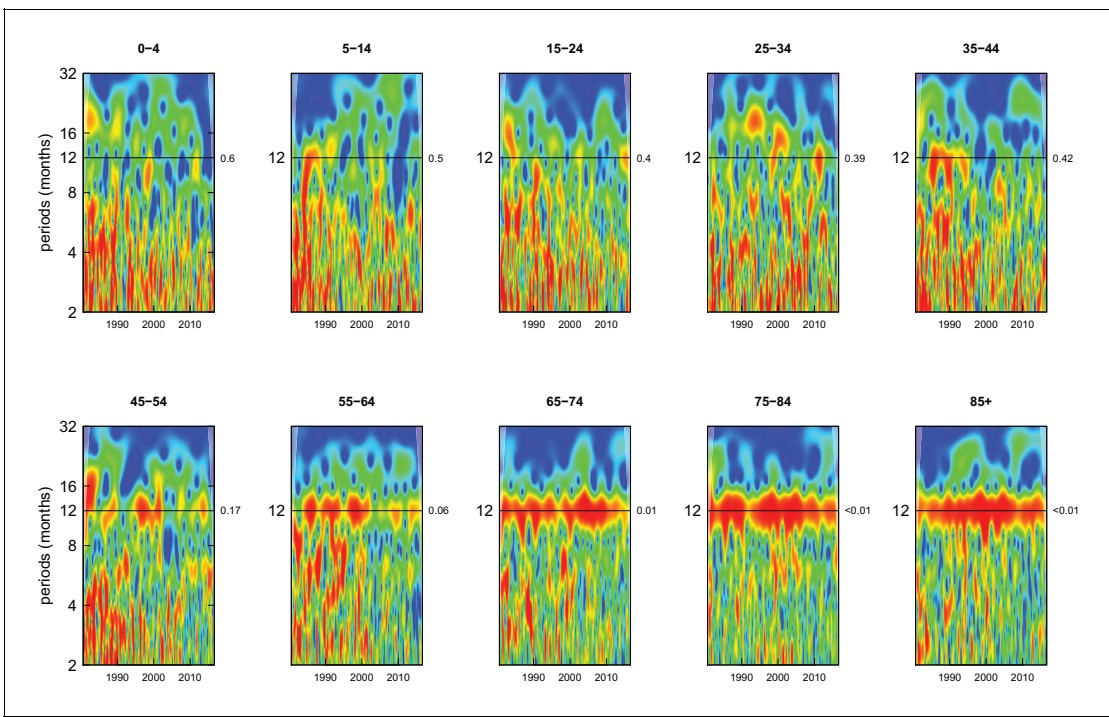

**Figure 4.** Wavelet power spectra for national time series of cancer death rates for 1980–2016, by age group for females. Wavelet power values increase from blue to red. The shaded regions at the left and right edge of each box indicate the cone of influence, where spectral analysis is less robust. P-values for the presence of 12 month seasonality are to the right of each figure at the 12 month line.

DOI: https://doi.org/10.7554/eLife.35500.007

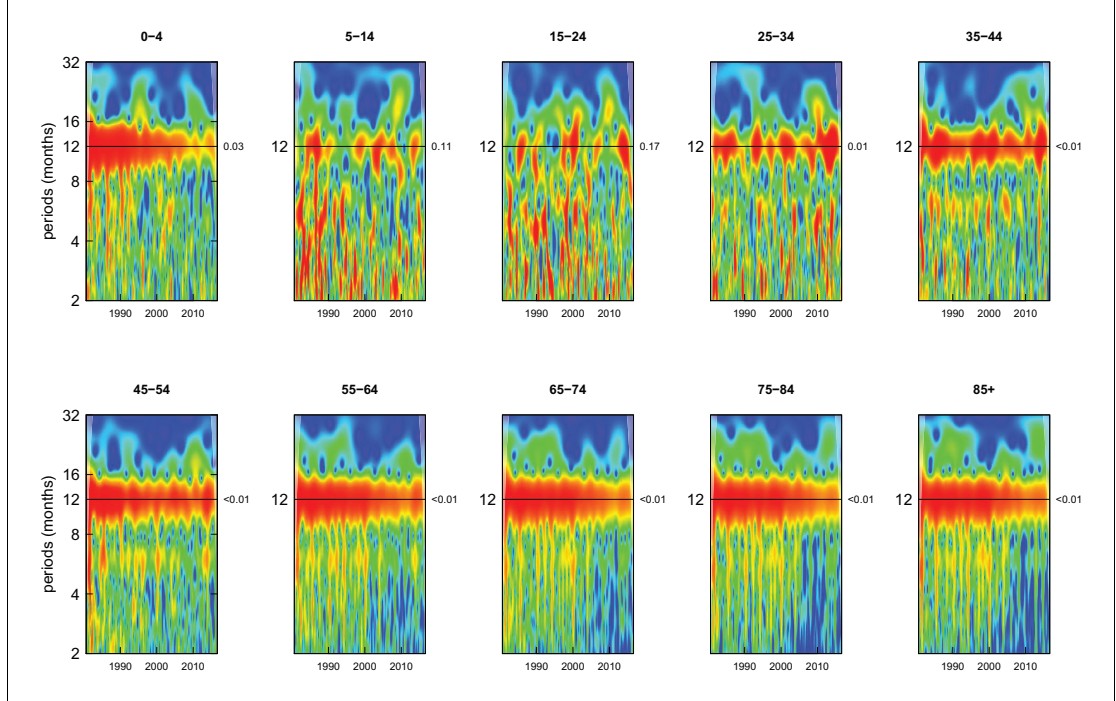

**Figure 5.** Wavelet power spectra for national time series of cardiorespiratory disease death rates for 1980–2016, by age group for males. Wavelet power values increase from blue to red. The shaded regions at the left and right edge of each box indicate the cone of influence, where spectral analysis is less robust. P-values for the presence of 12 month seasonality are to the right of each figure at the 12 month line.
DOI: https://doi.org/10.7554/eLife.35500.008

The following figure supplements are available for figure 5:

**Figure supplement 1.** Wavelet power spectra for national time series of cardiovascular disease death rates for 1980–2016, by age group for males.
DOI: https://doi.org/10.7554/eLife.35500.009
**Figure supplement 2.** Wavelet power spectra for national time series of chronic respiratory disease death rates for 1980–2016, by age group for males.
DOI: https://doi.org/10.7554/eLife.35500.010
**Figure supplement 3.** Wavelet power spectra for national time series of respiratory infection death rates for 1980–2016, by age group for males.
DOI: https://doi.org/10.7554/eLife.35500.011

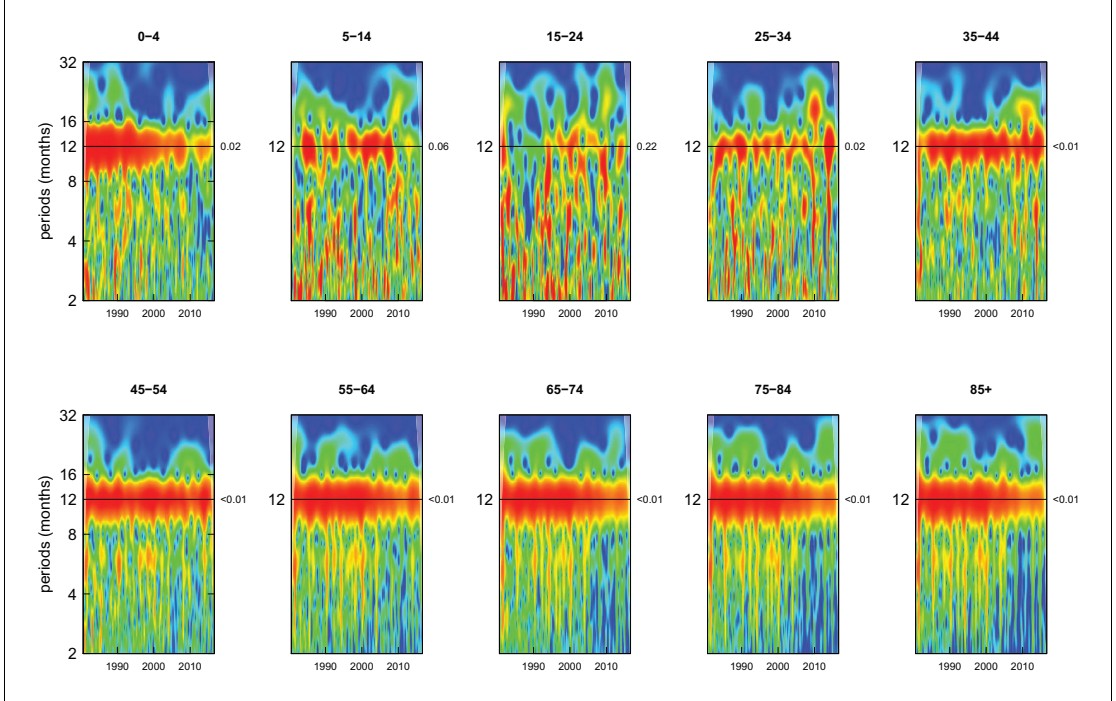

**Figure 6.** Wavelet power spectra for national time series of cardiorespiratory disease death rates for 1980–2016, by age group for females. Wavelet power values increase from blue to red. The shaded regions at the left and right edge of each box indicate the cone of influence, where spectral analysis is less robust. P-values for the presence of 12 month seasonality are to the right of each figure at the 12 month line.
DOI: https://doi.org/10.7554/eLife.35500.012

The following figure supplements are available for figure 6:

**Figure supplement 1.** Wavelet power spectra for national time series of cardiovascular disease death rates for 1980–2016, by age group for females.
DOI: https://doi.org/10.7554/eLife.35500.013

**Figure supplement 2.** Wavelet power spectra for national time series of chronic respiratory disease death rates for 1980–2016, by age group for females.
DOI: https://doi.org/10.7554/eLife.35500.014

**Figure supplement 3.** Wavelet power spectra for national time series of respiratory infection death rates for 1980–2016, by age group for females.
DOI: https://doi.org/10.7554/eLife.35500.015

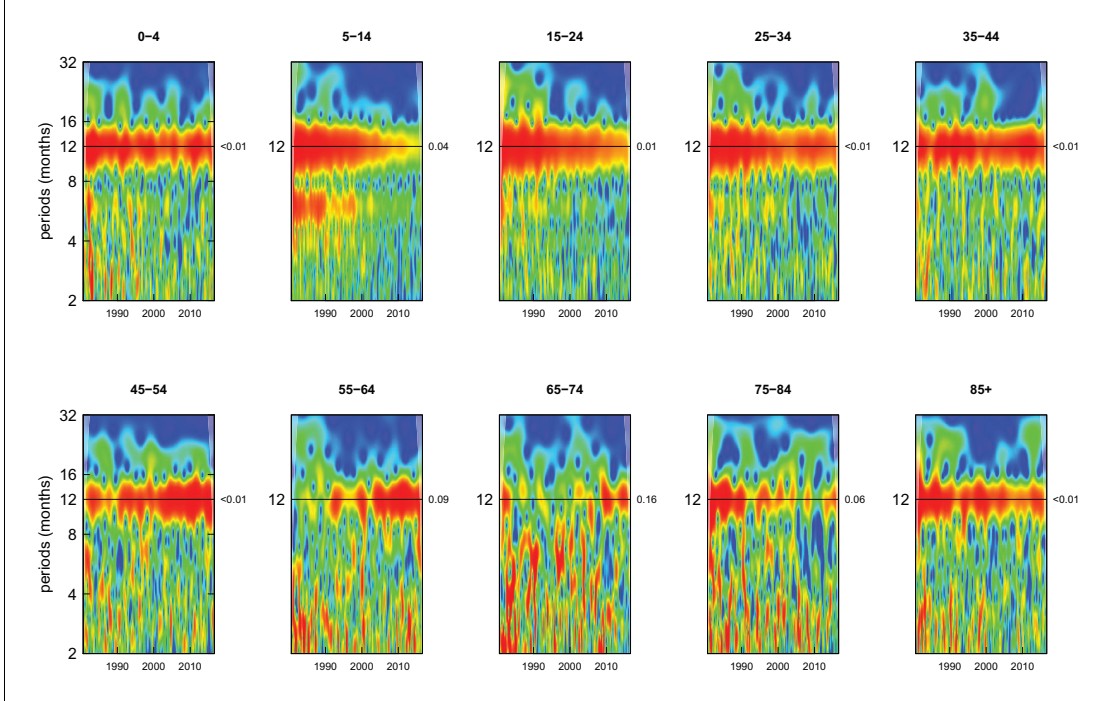

**Figure 7.** Wavelet power spectra for national time series of injury death rates for 1980–2016, by age group for males. Wavelet power values increase from blue to red. The shaded regions at the left and right edge of each box indicate the cone of influence, where spectral analysis is less robust. P-values for the presence of 12 month seasonality are to the right of each figure at the 12 month line.

DOI: https://doi.org/10.7554/eLife.35500.016

The following figure supplements are available for figure 7:

**Figure supplement 1.** Wavelet power spectra for national time series of intentional injury death rates for 1980–2016, by age group for males.
DOI: https://doi.org/10.7554/eLife.35500.017
**Figure supplement 2.** Wavelet power spectra for national time series of unintentional injury death rates for 1980–2016, by age group for males.
DOI: https://doi.org/10.7554/eLife.35500.018

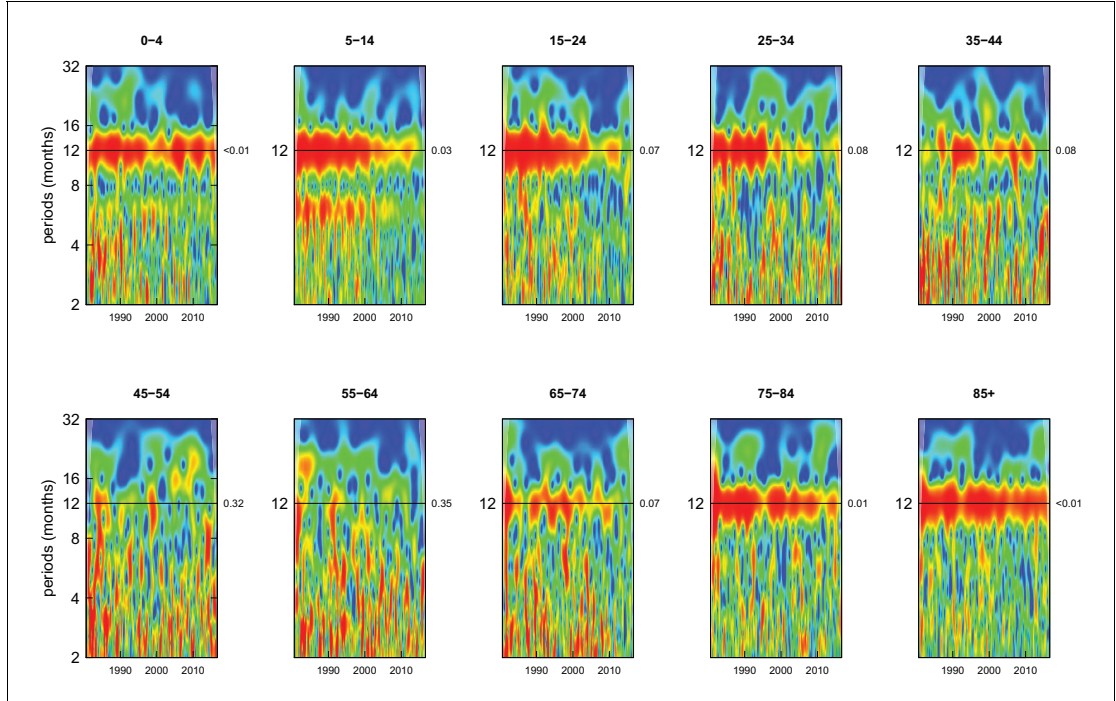

**Figure 8.** Wavelet power spectra for national time series of injury death rates for 1980–2016, by age group for females. Wavelet power values increase from blue to red. The shaded regions at the left and right edge of each box indicate the cone of influence, where spectral analysis is less robust. P-values for the presence of 12 month seasonality are to the right of each figure at the 12 month line.

DOI: https://doi.org/10.7554/eLife.35500.019

The following figure supplements are available for figure 8:

**Figure supplement 1.** Wavelet power spectra for national time series of intentional injury death rates for 1980–2016, by age group for females.
DOI: https://doi.org/10.7554/eLife.35500.020

**Figure supplement 2.** Wavelet power spectra for national time series of unintentional injury death rates for 1980–2016, by age group for females.
DOI: https://doi.org/10.7554/eLife.35500.021

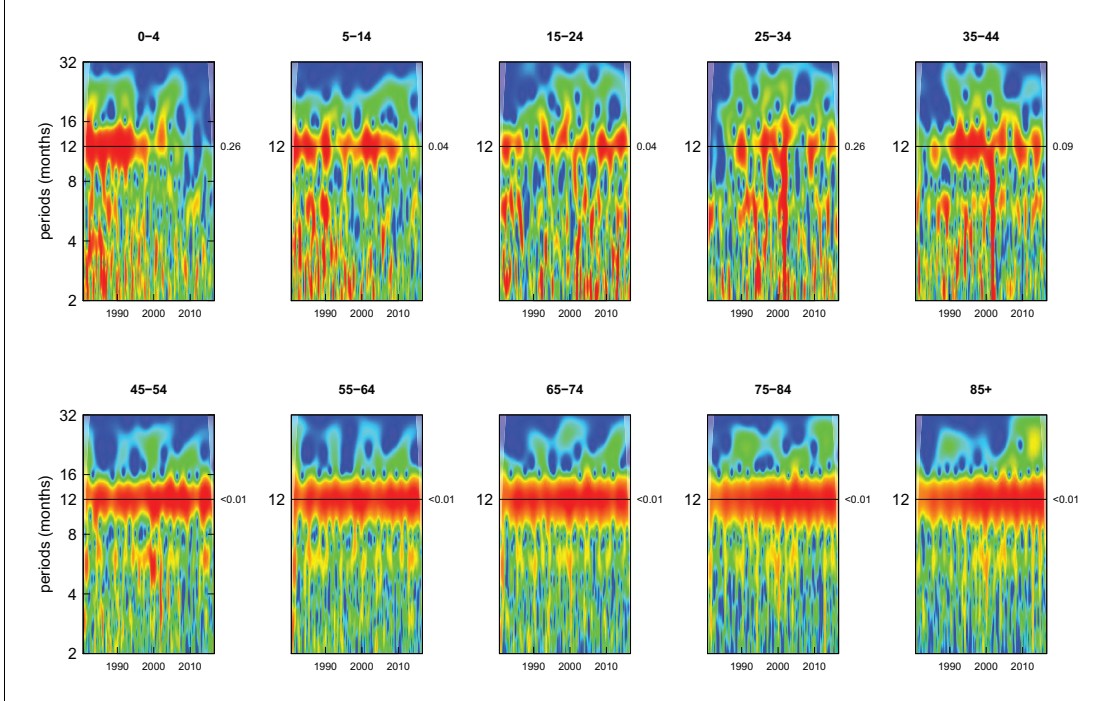

**Figure 9.** Wavelet power spectra for national time series of death rates from causes other than cancers, cardiorespiratory diseases and injuries for 1980–2016, by age group for males. Wavelet power values increase from blue to red. The shaded regions at the left and right edge of each box indicate the cone of influence, where spectral analysis is less robust. P-values for the presence of 12 month seasonality are to the right of each figure at the 12 month line.

DOI: https://doi.org/10.7554/eLife.35500.022

The following figure supplements are available for figure 9:

**Figure supplement 1.** Wavelet power spectra for national time series of substance use disorder death rates for 1980–2016, by age group for males.
DOI: https://doi.org/10.7554/eLife.35500.023

**Figure supplement 2.** Wavelet power spectra for national time series of perinatal condition death rates for 1980–2016, by age group for males.
DOI: https://doi.org/10.7554/eLife.35500.024

**Figure supplement 3.** Wavelet power spectra for national time series of endocrine disorder death rates for 1980–2016, by age group for males.
DOI: https://doi.org/10.7554/eLife.35500.025

**Figure supplement 4.** Wavelet power spectra for national time series of genitourinary disease death rates for 1980–2016, by age group for males.
DOI: https://doi.org/10.7554/eLife.35500.026

**Figure supplement 5.** Wavelet power spectra for national time series of neuropsychiatric disorder death rates for 1980–2016, by age group for males.
DOI: https://doi.org/10.7554/eLife.35500.027

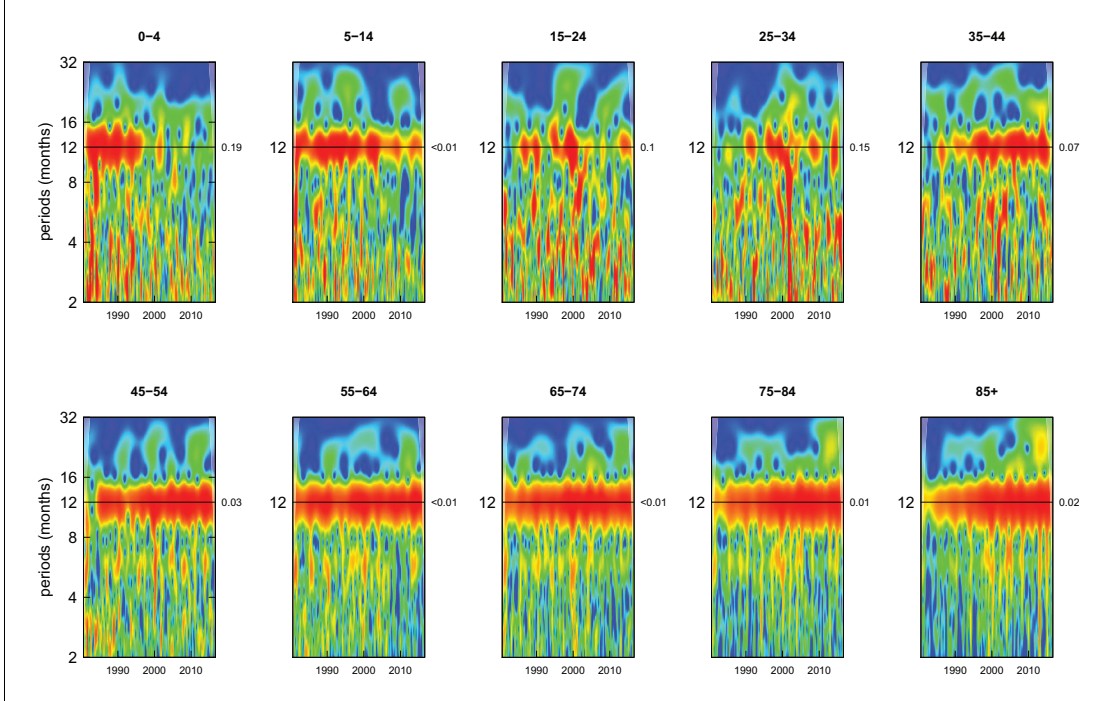

**Figure 10.** Wavelet power spectra for national time series of death rates from causes other than cancers, cardiorespiratory diseases and injuries for 1980–2016, by age group for females. Wavelet power values increase from blue to red. The shaded regions at the left and right edge of each box indicate the cone of influence, where spectral analysis is less robust. P-values for the presence of 12 month seasonality are to the right of each figure at the 12 month line.

DOI: https://doi.org/10.7554/eLife.35500.028

The following figure supplements are available for figure 10:

**Figure supplement 1.** Wavelet power spectra for national time series of substance use disorder death rates for 1980–2016, by age group for females.
DOI: https://doi.org/10.7554/eLife.35500.029

**Figure supplement 2.** Wavelet power spectra for national time series of maternal condition death rates for 1980–2016, by age group for females.
DOI: https://doi.org/10.7554/eLife.35500.030

**Figure supplement 3.** Wavelet power spectra for national time series of perinatal condition death rates for 1980–2016, by age group for females.
DOI: https://doi.org/10.7554/eLife.35500.031

**Figure supplement 4.** Wavelet power spectra for national time series of endocrine disorder death rates for 1980–2016, by age group for females.
DOI: https://doi.org/10.7554/eLife.35500.032

**Figure supplement 5.** Wavelet power spectra for national time series of genitourinary disease death rates for 1980–2016, by age group for females.
DOI: https://doi.org/10.7554/eLife.35500.033

**Figure supplement 6.** Wavelet power spectra for national time series of neuropsychiatric disorder death rates for 1980–2016, by age group for females.
DOI: https://doi.org/10.7554/eLife.35500.034

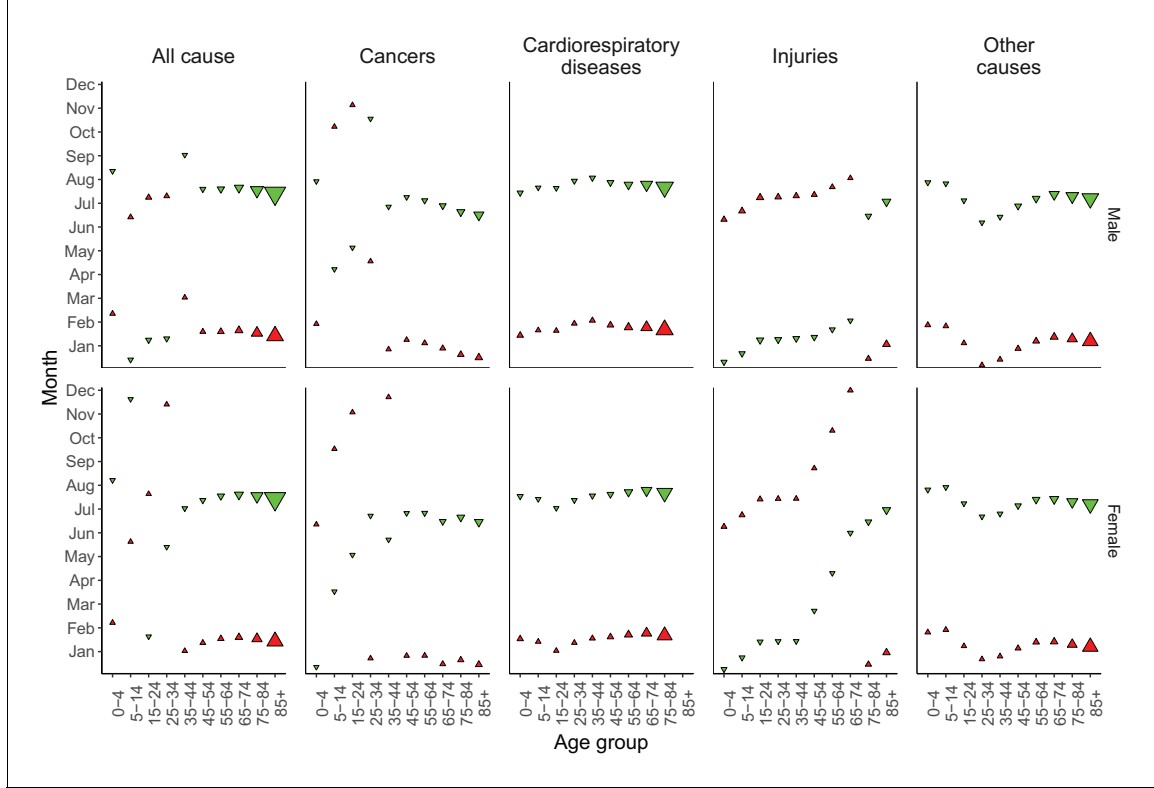

**Figure 11.** Mean timing of maximum and minimum all-cause and cause-specific mortality at the national level, by sex and age group for 1980–2016. Red arrows indicate the month of maximum mortality, and green arrows that of minimum mortality. The size of the arrow is inversely proportional to its respective variance.

DOI: https://doi.org/10.7554/eLife.35500.035

The following figure supplements are available for figure 11:

**Figure supplement 1.** Mean timing of maximum and minimum mortality for specific cardiorespiratory diseases at the national level, by sex and age group for 1980–2016.

DOI: https://doi.org/10.7554/eLife.35500.036

**Figure supplement 2.** Mean timing of maximum and minimum mortality for specific injuries at the national level, by sex and age group for 1980–2016.

DOI: https://doi.org/10.7554/eLife.35500.037

**Figure supplement 3.** Mean timing of maximum and minimum mortality for the cluster of causes other than cancers, cardiorespiratory diseases and injuries at the national level, by sex and age group for 1980–2016.

DOI: https://doi.org/10.7554/eLife.35500.038

Furthermore, above 45 years of age, there was little inter-region variation in the percent seasonal difference in all-cause mortality, despite the large variation in temperature difference between the peak and minimum months (*Figure 17*).

## Strengths and limitations

The strengths of our study are its innovative methods of characterizing seasonality of mortality dynamically over space and time, by age group and cause of death; using wavelet and centre of gravity analyses; using ERA-Interim data output to compare the association between seasonality of death rates and regional temperature. A limitation of our study is that we did not investigate seasonality of mortality by socioeconomic characteristics which may help with understanding its determinants and planning responses.

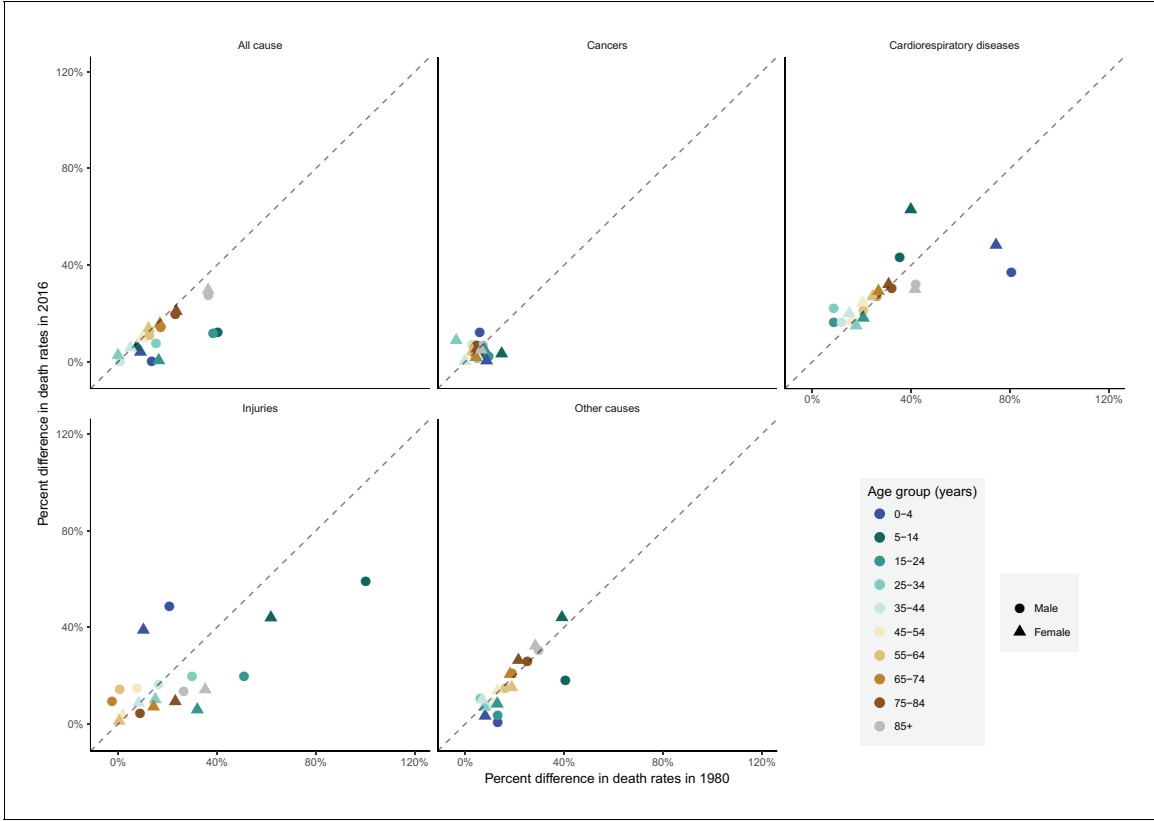

**Figure 12.** National percent difference in death rates between the maximum and minimum mortality months for all-cause and cause-specific mortality in 2016 versus 1980, by sex and age group.

DOI: https://doi.org/10.7554/eLife.35500.039

The following figure supplements are available for figure 12:

**Figure supplement 1.** National percent difference in death rates between the maximum and minimum mortality months for specific injuries in 2016 versus 1980, by sex and age group.

DOI: https://doi.org/10.7554/eLife.35500.040

**Figure supplement 2.** National percent difference in death rates between the maximum and minimum mortality months for specific cardiorespiratory diseases in 2016 versus 1980, by sex and age group.

DOI: https://doi.org/10.7554/eLife.35500.041

**Figure supplement 3.** National percent difference in death rates between the maximum and minimum mortality months for the cluster of causes other than cancers, cardiorespiratory diseases and injuries in 2016 versus 1980, by sex and age group.

DOI: https://doi.org/10.7554/eLife.35500.042

## Discussion

We used wavelet and centre of gravity analyses, which allowed systematically identifying and characterizing seasonality of total and cause-specific mortality in the USA, and examining how seasonality has changed over time. We identified distinct seasonal patterns in relation to age and sex, including higher all-cause summer mortality in young men (*Feinstein, 2002*; *Rau et al., 2018*). Importantly, we also showed that all-cause and cause-specific mortality seasonality is largely similar in terms of both timing and magnitude across diverse climatic regions with substantially different summer and winter temperatures. Insights of this kind would not have been possible analysing data averaged over time or nationally, or fixed to pre-specified frequencies.

Prior studies have noted seasonality of mortality for all-cause mortality and for specific causes of death in the USA (*Feinstein, 2002*; *Kalkstein, 2013*; *Rau, 2004*; *Rau et al., 2018*; *Rosenwaike, 1966*; *Seretakis et al., 1997*). Few of these studies have done consistent national and subnational analyses, and none has done so over time, for a comprehensive set of age groups and

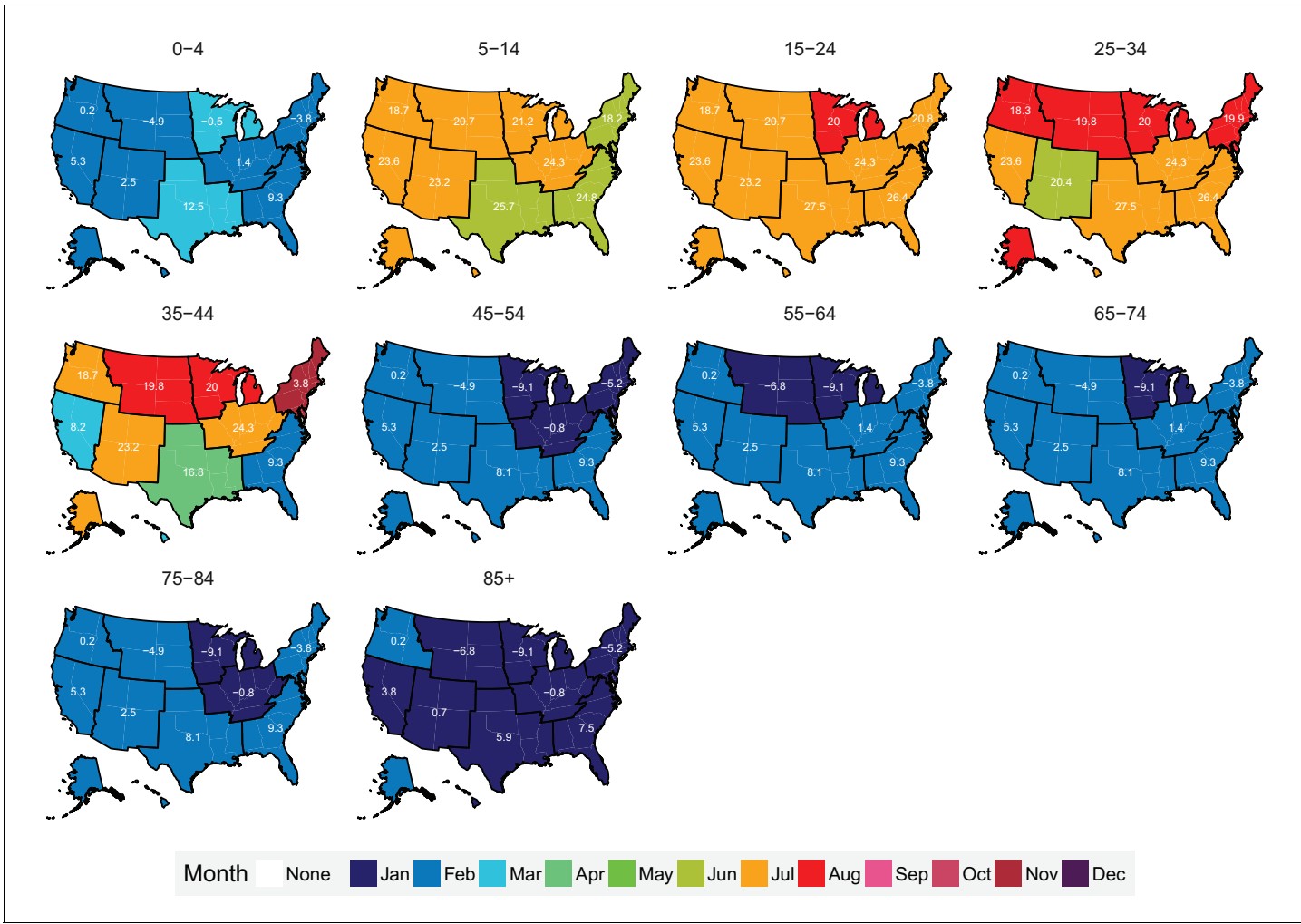

**Figure 13.** Mean timing of maximum all-cause mortality for 1980–2016, by climate region and age group for males. Average temperatures (in degrees Celsius) are included in white for the corresponding month of maximum and minimum mortality for each climate region.
DOI: https://doi.org/10.7554/eLife.35500.043

causes of death, and in relation to regional temperature differences. Our results on strong seasonality of cardiorespiratory diseases deaths and weak seasonality of cancer deaths, restricted to older ages, are broadly consistent with these studies (*Feinstein, 2002*; *Rau et al., 2018*; *Rosenwaike, 1966*; *Seretakis et al., 1997*), which had limited analysis on how seasonality changes over time and geography (*Feinstein, 2002*; *Rau et al., 2018*; *Rosenwaike, 1966*). Similarly, our results on seasonality of injury deaths are supported by a few prior studies (*Feinstein, 2002*; *Rau et al., 2018*; *Rosenwaike, 1966*), but our subnational analysis over three decades revealed variations in when injury deaths peaked and in how seasonal differences in these deaths have changed over time in relation to age group which had not been reported before.

A study of 36 cities in the USA, aggregated across age groups and over time, also found that excess mortality was not associated with seasonal temperature range (*Kinney et al., 2015*). In contrast, a European study found that the difference between winter and summer mortality was lower in colder Nordic countries than in warmer southern European nations (*Healy, 2003*; *McKee, 1989*) (the study's measure of temperature was mean annual temperature which differed from the temperature difference between maximum and minimum mortality used in our analysis although the two measures are correlated). The absence of variation in the magnitude of mortality seasonality indicates that different regions in the USA are similarly adapted to temperature

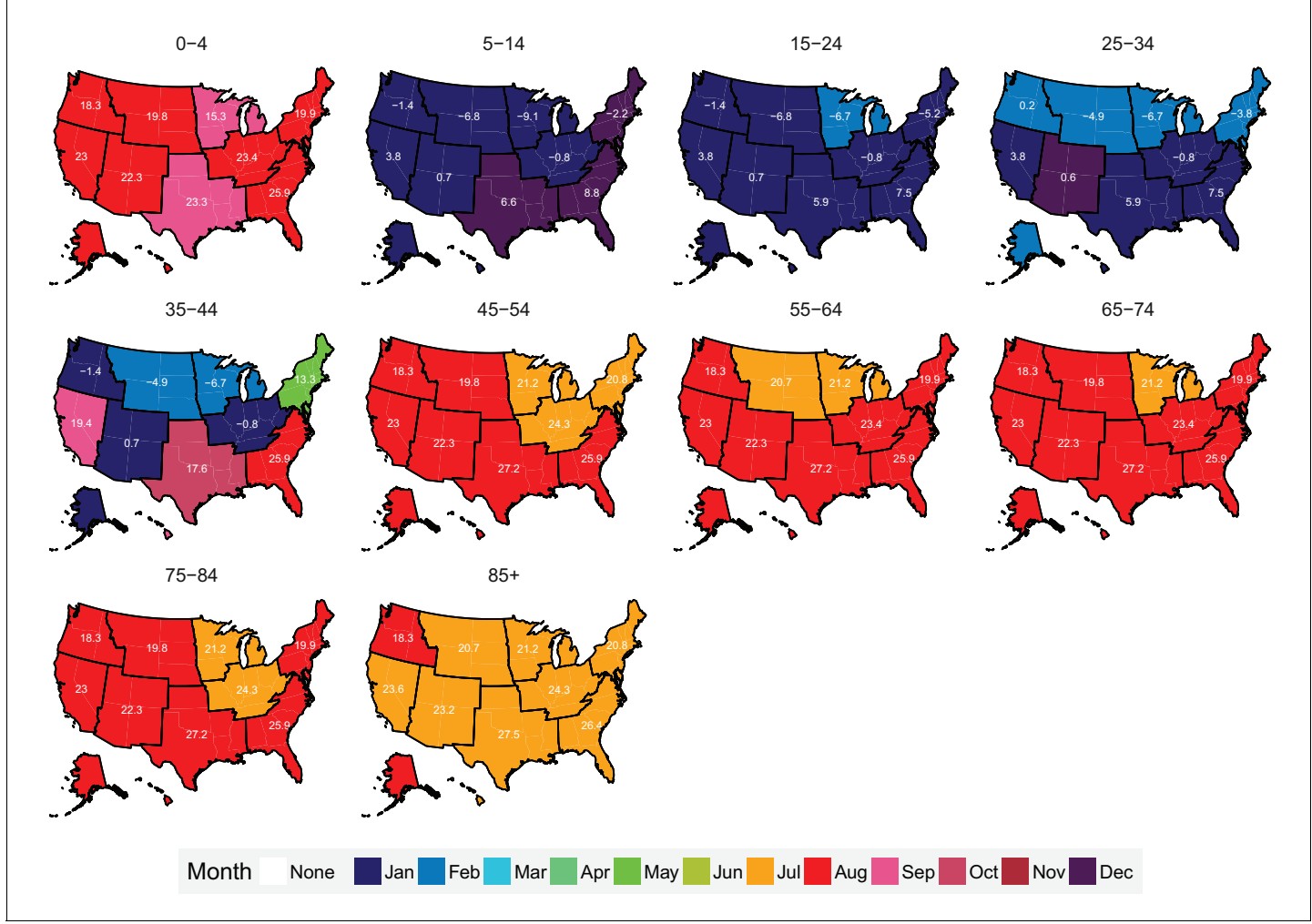

**Figure 14.** Mean timing of minimum all-cause mortality for 1980–2016, by climate region and age group for males. Average temperatures (in degrees Celsius) are included in white for the corresponding month of maximum and minimum mortality for each climate region.
DOI: https://doi.org/10.7554/eLife.35500.044

seasonality, whereas Nordic countries may have better environmental (e.g. housing insulation and heating) and health system measures to counter the effects of cold winters than those in southern Europe. If the observed absence of association between the magnitude of mortality seasonality and seasonal temperature difference across the climate regions also persists over time, the changes in temperature as a result of global climate change are unlikely to affect the winter-summer mortality difference.

The cause-specific analysis showed that the substantial decline in seasonal mortality differences in adolescents and young adults was related to the diminishing seasonality of (unintentional) injuries, especially from road traffic crashes, which are more likely to occur in the summer months (*Liu et al., 2005*) and are more common in men. The weakening of seasonality in boys under five years of age was related to two phenomena: first, the seasonality of death from cardiorespiratory diseases declined, and second, the proportion of deaths from perinatal conditions, which exhibit limited seasonality (*Figure 9—figure supplement 2* and *Figure 10—figure supplement 3*), increased (*MacDorman and Gregory, 2015*).

In contrast to young and middle ages, mortality in older ages, where death rates are highest, maintained persistent seasonality over a period of three decades (we note that although the percent seasonal difference in mortality has remained largely unchanged in these ages, the absolute

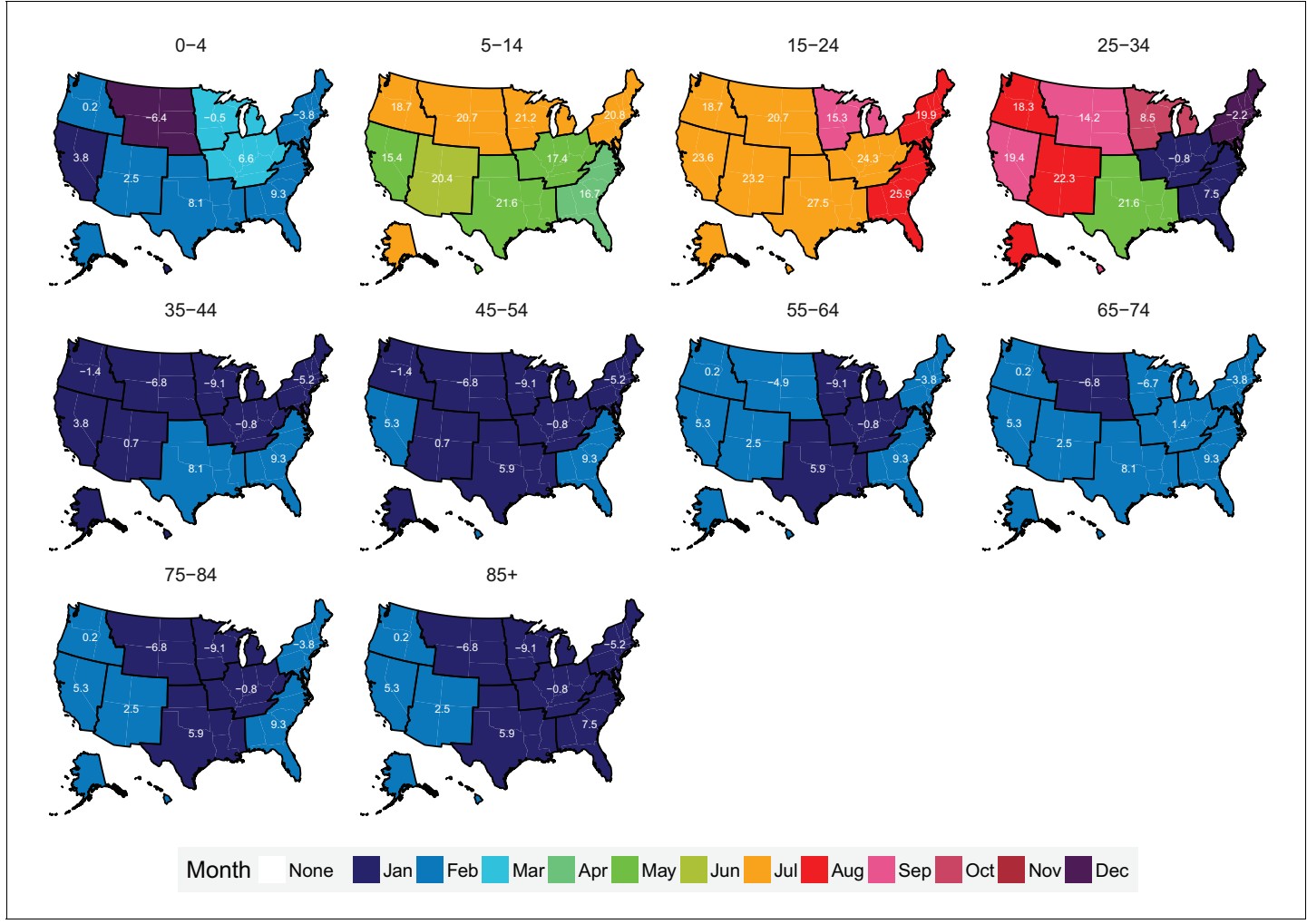

**Figure 15.** Mean timing of maximum all-cause mortality for 1980–2016, by climate region and age group for females. Average temperatures (in degrees Celsius) are included in white for the corresponding month of maximum and minimum mortality for each climate region.
DOI: https://doi.org/10.7554/eLife.35500.045

difference in death rates between the peak and minimum months has declined because total mortality has a declining long-term trend). This finding demonstrates the need for environmental and health service interventions targeted towards this group irrespective of geography and local climate. Examples of such interventions include enhancing the availability of both environmental and medical protective factors, such as better insulation of homes, winter heating provision and flu vaccinations, for the vulnerable older population (*Katiyo et al., 2017*). Social interventions, including regular visits to the isolated elderly during peak mortality periods to ensure that they are optimally prepared for adverse conditions, and responsive and high-quality emergency care, are also important to protect this vulnerable group (*Healy, 2003*; *Lerchl, 1998*; *Katiyo et al., 2017*). Emergent new technologies, such as always-connected hands-free communications devices with the outside world, in-house cameras, and personal sensors also provide an opportunity to enhance care for the older, more vulnerable groups in the population, especially in winter when the elderly have fewer social interactions (*Morris, 2013*). Such interventions are important today, and will remain so as the population ages and climate change increases the within- and between-season weather variability.

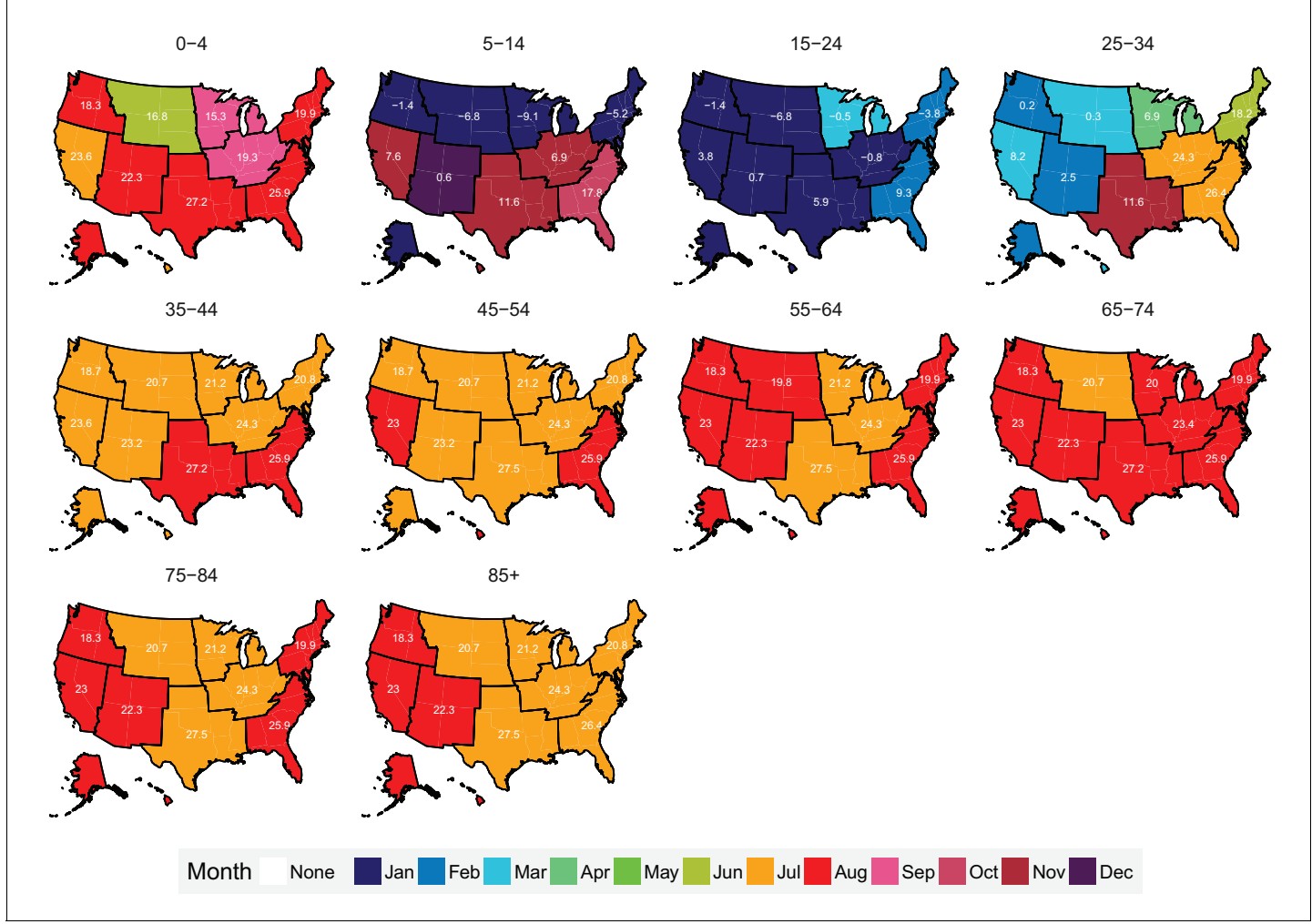

**Figure 16.** Mean timing of minimum all-cause mortality for 1980–2016, by climate region and age group for females. Average temperatures (in degrees Celsius) are included in white for the corresponding month of maximum and minimum mortality for each climate region.
DOI: https://doi.org/10.7554/eLife.35500.046

## Materials and methods

### Data

We used data on all 85,854,176 deaths in the USA from 1980 to 2016 from the National Center for Health Statistics (NCHS). Age, sex, state of residence, month of death, and underlying cause of death were available for each record. The underlying cause of death was coded according to the international classification of diseases (ICD) system (9[th] revision of ICD from 1980 to 1998 and 10[th] revision of ICD thereafter). Yearly population counts were available from NCHS for 1990 to 2016 and from the US Census Bureau prior to 1990 (*Ingram et al., 2003*). We calculated monthly population counts through linear interpolation, assigning each yearly count to July.

We also subdivided the national data geographically into nine climate regions used by the National Oceanic and Atmospheric Administration (*Figure 18* and *Table 2*) (*Karl and Koss, 1984*). On average, the Southeast and South are the hottest climate regions with average annual temperatures of 18.4°C and 18°C respectively; the South also possesses the highest average maximum monthly temperature (27.9°C in July). The lowest variation in temperature throughout the year is that of the Southeast (an average range of 17.5°C). The three coldest climate regions are West North Central, East North Central and the Northwest (7.6°C, 8.0°C, 8.2°C respectively). Mirroring the characteristics of the hottest climate regions, the largest variation in temperature throughout the year is

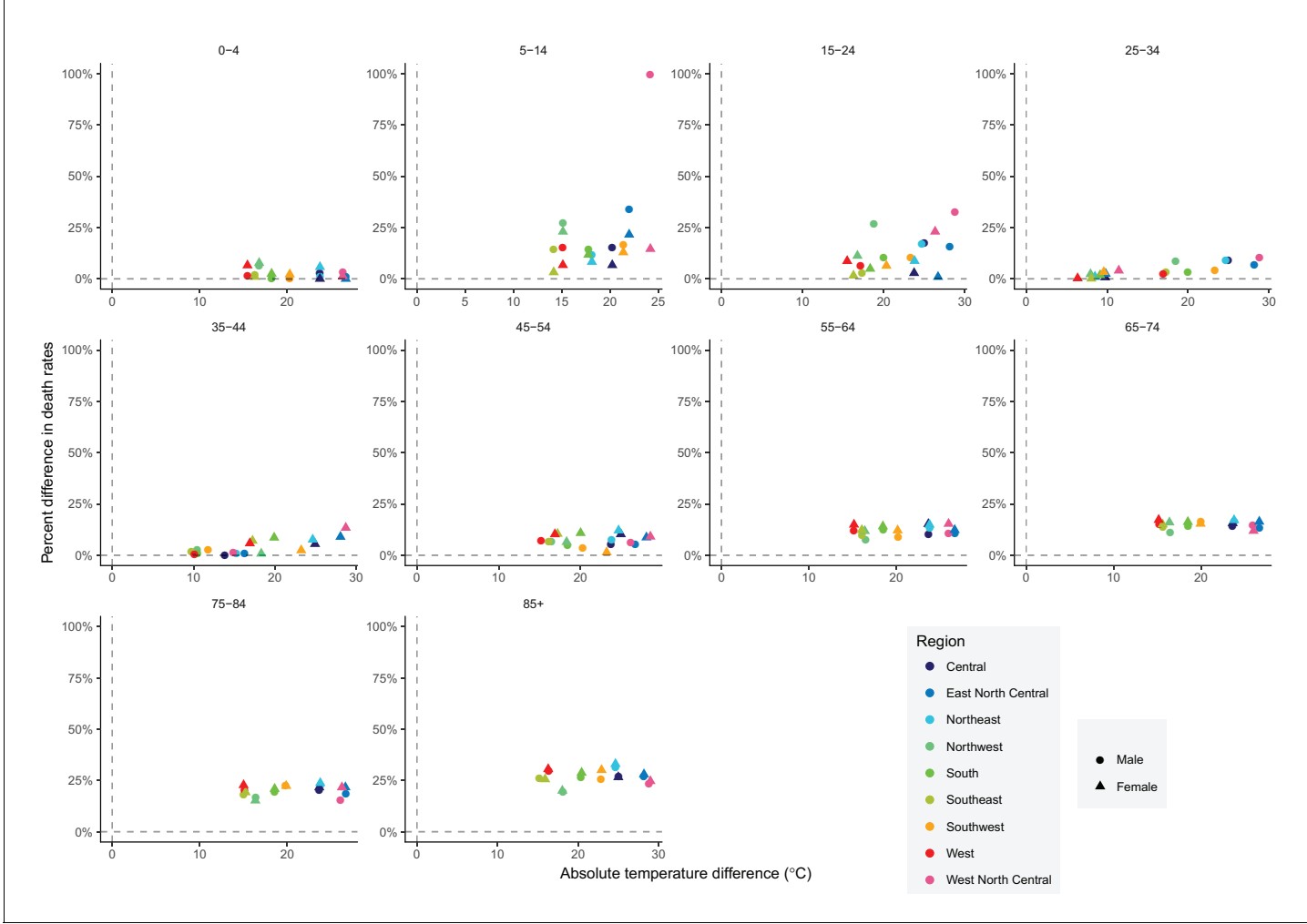

**Figure 17.** The relationship between percent difference in all-cause death rates and temperature difference between months with maximum and minimum mortality across climate regions, by sex and age group in 2016.

DOI: https://doi.org/10.7554/eLife.35500.047

that of the coldest region, West North Central (an average range of 30.5°C), which also has the lowest average minimum monthly temperature (−6.5°C in January). The other climate regions, Northeast, Southwest, and Central, possess similar average temperatures (10°C to 14°C) and variation within the year of (23°C to 26°C), with the Northeast being the most populous region in the United States (with 19.8% total population in 2016).

Data were divided by sex and age in the following 10 age groups: 0–4, 5–14, 15-24, 25–34, 35–44, 45–54, 55–64, 65–74, 75–84, 85+ years. We calculated monthly death rates for each age and sex group, both nationally and for sub-national climate regions. Death rate calculations accounted for varying length of months, by multiplying each month's death count by a factor that would make it equivalent to a 31 day month.

For analysis of seasonality by cause of death, we mapped each ICD-9 and ICD-10 codes to four main disease categories (*Table 1*) and to a number of subcategories which are presented in the Supplementary Note. Cardiorespiratory diseases and cancers accounted for 56.4% and 21.2% of all deaths in the USA, respectively, in 1980, and 40.3% and 22.4%, respectively, in 2016. Deaths from cardiorespiratory diseases have been associated with cold and warm temperatures (*Basu, 2009*; *Basu and Samet, 2002*; *Bennett et al., 2014*; *Braga et al., 2002*; *Gasparrini et al., 2015*). Injuries,

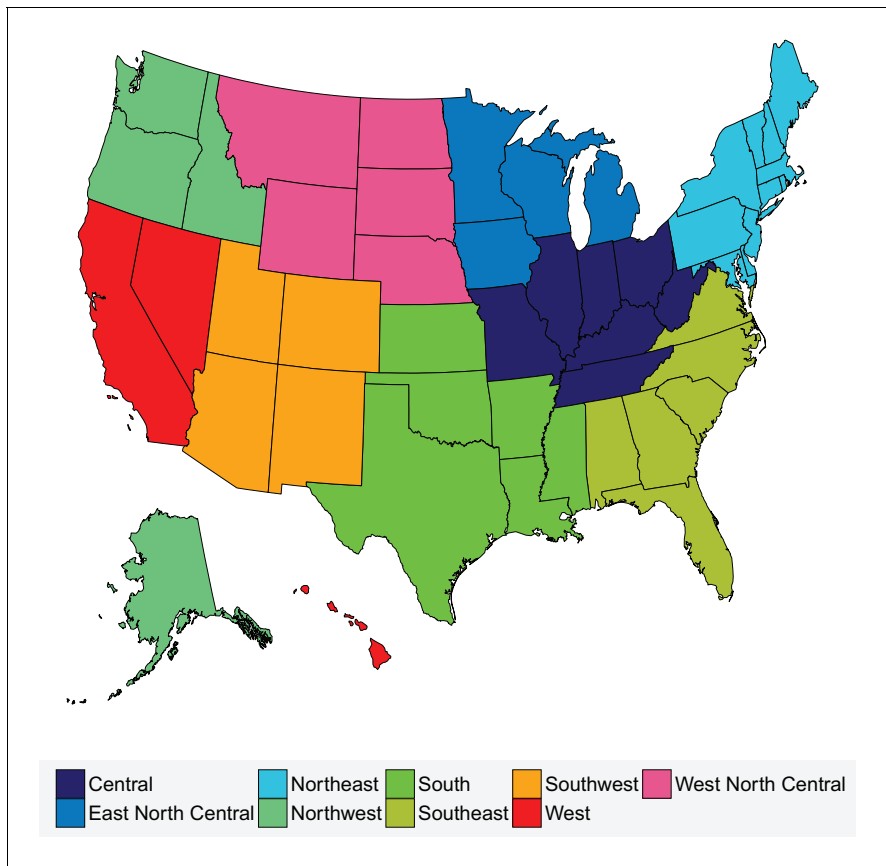

Central
East North Central
Northeast
Northwest
South
Southeast
Southwest
West
West North Central

**Figure 18.** Climate regions of the USA.
DOI: https://doi.org/10.7554/eLife.35500.048

**Table 2.** Characteristics of climate regions of the USA.

| Climate region | Constituent states | Population (2016) | Mean annual temperature (1980–2016) (°C) |
|---|---|---|---|
| Central | Illinois, Indiana, Kentucky, Missouri, Ohio, Tennessee, West Virginia | 50,191,326 | 11.6 |
| East North Central | Iowa, Michigan, Minnesota, Wisconsin | 24,418,738 | 8 |
| Northeast | Connecticut, Delaware, Maine, Maryland, Massachusetts, New Hampshire, New Jersey, New York, Pennsylvania, Rhode Island, Vermont | 64,046,741 | 10.6 |
| Northwest | Alaska, Idaho, Oregon, Washington | 13,811,810 | 8.2 |
| South | Arkansas, Kansas, Louisiana, Mississippi, Oklahoma, Texas | 45,388,414 | 18 |
| Southeast | Alabama, Florida, Georgia, North Carolina, South Carolina, Virginia | 59,356,072 | 18.4 |
| Southwest | Arizona, Colorado, New Mexico, Utah | 17,613,981 | 13.6 |
| West | California, Hawaii, Nevada | 43,708,574 | 16.6 |
| West North Central | Montana, Nebraska, North Dakota, South Dakota, Wyoming | 5,168,753 | 7.6 |

DOI: https://doi.org/10.7554/eLife.35500.049

which accounted for 8% of all deaths in the USA in 1980 and 7.3% in 2016, may have seasonality that is distinct from so-called natural causes. We did not further divide other causes because the number of deaths could become too small to allow stable estimates when divided by age group, sex and climate region.

We obtained data on temperature from ERA-Interim, which combines predictions from a physical model with ground-based and satellite measurements (*Dee et al., 2011*). We used gridded four-times-daily estimates at a resolution of 80 km to generate monthly population-weighted temperature by climate region throughout the analysis period.

## Statistical methods

We used wavelet analysis to investigate seasonality for each age-sex group. Wavelet analysis uncovers the presence, and frequency, of repeated maxima and minima in each age-sex-specific death rate time series (*Hubbard, 1998*; *Torrence and Compo, 1998*). In brief, a Morlet wavelet, described in detail elsewhere (*Cazelles et al., 2008*), is equivalent to using a moving window on the death rate time series and analysing periodicity in each window using a short-form Fourier transform, hence generating a dynamic spectral analysis, which allows measuring dynamic seasonal patterns, in which the periodicity of death rates may disappear, emerge, or change over time. In addition to coefficients that measure the frequency of periodicity, wavelet analysis estimates the probability of whether the data are different from the null situation of random fluctuations that can be represented with white (an independent random process) or red (autoregressive of order one process) noise. For each age-sex group, we calculated the p-values of the presence of 12 month seasonality for the comparison of wavelet power spectra of the entire study period (1980–2016) with 100 simulations against a white noise spectrum, which represents random fluctuations. We used the R package WaveletComp (version 1.0) for the wavelet analysis. Before analysis, we de-trended death rates using a polynomial regression, and rescaled each death rate time series so as to range between 1 and −1.

To identify the months of maximum and minimum death rates, we calculated the centre of gravity and the negative centre of gravity of monthly death rates. Centre of gravity was calculated as a weighted average of months of deaths, with each month weighted by its death rate; negative centre of gravity was also calculated as a weighted average of months of deaths, but with each month was weighted by the difference between its death rate and the year's maximum death rate. In taking the weighted average, we allowed December (month 12) to neighbour January (month 1), representing each month by an angle subtended from 12 equally-spaced points around a unit circle. Using a technique called circular statistics, a mean $\left(\bar{\theta}\right)$ of the angles $(\theta_1, \theta_2, \theta_3 \ldots, \theta_{n,})$ representing the deaths (with n the total number of deaths in an age-sex group for a particular cause of death) is found using the relation below:

$$\bar{\theta} = arg\left\{\sum_{j=1}^{n}\exp\left(\mathrm{i}\theta_j\right)\right\},$$

where *arg* denotes the complex number argument and $\theta_j$ denotes the month of death in angular form for a particular death *j*. The outcome of this calculation is then converted back into a month value (*Fisher, 1995*). Along with each circular mean, a 95% confidence interval (CI) was calculated by using 1000 bootstrap samples. The R package CircStats (version 0.2.4) was used for this analysis.

For each age-sex group and cause of death, and for each year, we calculated the percent difference in death rates between the maximum and minimum mortality months. We fitted a linear regression to the time series of seasonal differences from 1980 to 2016, and used the fitted trend line to estimate how much the percentage difference in death rates between the maximum and minimum mortality months had changed from 1980 to 2016. We weighted seasonal difference by the inverse of the square of its standard error, which was calculated using a Poisson model to take population size of each age-sex group through time into account. This method gives us a p-value for the change in seasonal difference per year, which we used to calculate the seasonal difference at the start (1980) and end (2016) of the period of study. Our method of analysing seasonal differences avoids assuming that any specific month or group of months represent highest and lowest number of deaths for a particular cause of death, which is the approach taken by the traditional measure of Excess Winter

Deaths. It also allows the maximum and minimum mortality months to vary by age group, sex and cause of death.

## Acknowledgments

Robbie Parks is supported by a Wellcome Trust ISSF Studentship. Work on the US mortality data is supported by a grant from US Environmental Protection Agency. The views expressed in this document are solely those of the authors and do not necessarily reflect those of the Agency.

## Additional information

### Competing interests

Majid Ezzati: Reports a charitable grant from the AstraZeneca Young Health Programme, and personal fees from Prudential, Scor, and Third Bridge, all outside the submitted work. The other authors declare that no competing interests exist.

### Funding

| Funder | Grant reference number | Author |
|---|---|---|
| Wellcome Trust | 205208/Z/16/Z | Majid Ezzati |
| U.S. Environmental Protection Agency | RD-83587301 | Majid Ezzati |

The funders had no role in study design, data collection and interpretation, or the decision to submit the work for publication.

### Author contributions

Robbie M Parks, Conceptualization, Data curation, Software, Formal analysis, Investigation, Visualization, Methodology, Writing—original draft; James E Bennett, Conceptualization, Visualization, Methodology, Writing—review and editing; Kyle J Foreman, Data curation, Methodology, Writing—review and editing; Ralf Toumi, Conceptualization, Methodology, Writing—review and editing; Majid Ezzati, Conceptualization, Resources, Data curation, Supervision, Funding acquisition, Visualization, Methodology, Writing—original draft, Project administration

### Author ORCIDs

Robbie M Parks http://orcid.org/0000-0002-7916-1717
Majid Ezzati https://orcid.org/0000-0002-2109-8081

### Decision letter and Author response

Decision letter https://doi.org/10.7554/eLife.35500.062
Author response https://doi.org/10.7554/eLife.35500.063

## Additional files

### Supplementary files

• Transparent reporting form
DOI: https://doi.org/10.7554/eLife.35500.050

### Data availability

The ERA-Interim temperature data are available at https://www.ecmwf.int/en/forecasts/datasets/reanalysis-datasets/era-interim. The US Census Populations With Bridged Race Categories data for 1990-2016 are available at https://wonder.cdc.gov/bridged-race-population.html. Pre 1990, the County Intercensal Tables are available at https://www.census.gov/data/tables/time-series/demo/popest/1980s-county.html. The Vital statistics data are available at https://www.cdc.gov/nchs/nvss/dvs_data_release.htm through a request to NAPHSIS (https://www.naphsis.org/).

The following previously published datasets were used:

| Author(s) | Year | Dataset title | Dataset URL | Database and Identifier |
|---|---|---|---|---|
| National Center for Health Statistics | 2017 | Vital statistics (1980-2016) | https://www.cdc.gov/nchs/nvss/dvs_data_release.htm | National Center for Health Statistics, dvs_data_release |
| European Centre for Medium-Range Weather Forecasts | 2016 | ERA-Interim temperature data (1979-2016) | https://www.ecmwf.int/en/forecasts/datasets/reanalysis-datasets/era-interim | European Centre for Medium-Range Weather Forecasts, reanalysis-datasets/era-interim |
| CDC Wonder | 2016 | US Census Populations With Bridged Race Categories (1990-2016) | https://wonder.cdc.gov/bridged-race-population.html | CDC WONDER, bridged-race-population |
| United States Census Bureau | 2016 | County Intercensal Tables (1980-1989) | https://www.census.gov/data/tables/time-series/demo/popest/1980s-county.html | United States Census Bureau, 1980s-county |

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
