## [Decision Letter]

Thank you for submitting your article "National and regional seasonal dynamics of all-cause and cause-specific mortality in the USA from 1980 to 2013" for consideration by *eLife*. Your article has been reviewed by three peer reviewers, and the evaluation has been overseen by Eduardo Franco as a Reviewing Editor and Prabhat Jha as the Senior Editor. The Reviewing Editor has drafted this decision to help you prepare a revised submission. As is common in *eLife*, decision narratives are an amalgamation of the essential points of the reviewers' critiques after eliminating redundancies in comments and suggestions.

Summary:

This paper examines the seasonality in age- and disease-specific mortality in the US using geocoded data analyzed via wavelet statistical modelling and Poisson regression techniques. The key results indicate mortality increases in adults over 45 years old in winter from cardiorespiratory causes and injuries. Injuries, attributable mainly to road traffic crashes were seen to rise in the summer months among young men. No variation in seasonality was found by climate region, indicating no apparent disparities in adaptation by geographic location. This study provides insights as to whether climate changes could have an impact on the mortality patterns observed in the US as a model. The findings could assist in planning surveillance mechanisms and in projecting workforce requirements in primary and emergency healthcare.

Essential revisions:

1) By choosing only 4 disease groups the authors seem to have missed a key opportunity to investigate seasonal patterns in finer subgroups despite possessing a large sample size (n = 77,771,264), Furthermore, the rationale for choosing the subgroups presented is unclear. For example, the authors chose to group cardiorespiratory diseases into one large outcome when cardiovascular and respiratory outcomes have differing mechanisms in their association with heat or cold exposure. Within respiratory diseases itself, it makes sense to separate acute and chronic causes because the mechanisms vary so starkly and are related to seasonal patterns, i.e. the incidence of pneumonia and acute respiratory outcomes are likely heightened in the winter and this observation may differ to how/why the incidence of chronic respiratory deaths vary seasonally. Furthermore, the mechanisms for how heat is associated with elevated respiratory deaths is understudied and knowing more about seasonal patterns here would be a useful addition to the literature.

I believe it's important to investigate seasonal patterns of deaths from infectious disease, maternal and neonatal causes, endocrine disorders, genitourinary conditions and neuropsychiatric conditions (which are all associated with temperature) to add valuable insight into understudied but important disease groups, but these have been excluded from the present analysis. Considering the tragic opioid epidemic in USA, it would be a valuable contribution to the literature to understand whether seasonal patterns (at least in more recent years) are observed with deaths related to substance use disorders.

2) The authors pointed out in the Introduction that global warming may impact on excess cold weather death rates. Presumably a major part of the rationale for testing the differences in regions over time was to help inform our understanding of what the impact may be. However no conclusions were reached with regard to the implications of the findings. Clearly any such conclusions would need to be appropriately and heavily caveated but as this is a major reason for the analysis. Please expand the Discussion to include these points.

3) Methodology:

A) Wavelet power spectra are not always easy to interpret, and the uncertainty in estimated wavelet coefficients is difficult to quantify. The wavelet analysis makes for an interesting exploratory tool, showing that for the most part cycles have a duration of 12 months and are reasonably stable over time. It does not seem possible to draw firm conclusions about the research hypothesis using wavelets, however. All cause male mortality for 15-24 year olds appears to be less cyclical in recent years, although how to quantify this effect and assign a statistical significance to it is not apparent from the power spectrum shown.

The second analysis is more appropriate, although the details of this analysis are sparse. It appears that the model used is something like the following, where Yit is the count of deaths in age group i at time t.

-------------------------------------

Yit∼ Poisson(Nit ⋅ λit)log(λit)= μi + βit + Mit ⋅ [α1i ⋅ cos(2πt/12)

+α2i ⋅ sin(2πt/12)+α3i ⋅ cos(2πt/6)

+α4i ⋅ sin(2πt/6)]

Mit= ρi + γit

-------------------------------------

A model of this could answer the following questions, with p-values produced from a likelihood ratio test.

- Is γi negative? If so the seasonal effect is becoming less severe for age group i.

- Does it look like all age groups have the same trend, with γi = γ₀ for all i?

- Do all age groups have the same cycle, with αpi = αp0?

- Does γ vary by region (negative in cold-climate regions?)

- Do the cycles, given by the α, vary by region or is a nation-wide cycle sufficient?

The above model is easier to interpret than the wavelet analysis. The quantity 1 − exp(120γ) is the change per decade in the seasonality effect, which could be reported with a 95% confidence interval.

B) The approach is different to a number of those referenced (particularly the standard measure of excess winter deaths). It would be helpful to have some comment on this, particularly the generalisability to other studies. The focus has been on peak months in this analysis, but the standard assessment of excess winter deaths is to compare December to March months to the rest of the year.

C) Appropriate rationale was given to the regional splits used in the analysis. However no description was given of the characteristics of these regional areas other than subsequent information on temperatures. As pointed out there could be a number of factors that are important. I would expect some reference to their differing characteristics.

4) Interpretation:

A) No table is presented with descriptive statistics, i.e. on the sample size of cases that fall into each disease category.

B) I may be misreading the paper (in which case it would be helpful to clarify so others do not make the same mistake), however the analysis in Figure 6 compares the difference in temperature experienced in those regions between the warmest and coldest months versus the% seasonal difference in death. In the Discussion this in contrasted with findings from Europe where countries with a more temperate winter have, paradoxically, higher rates of excess winter mortality. However this is not an appropriate direct comparison, regional extremes in variation were not looked at when comparisons between countries are made. This does not invalidate the comparison but it would be helpful to explicitly summarise if there are any differences between regions and examine what those are.

C) One option for this paper would be to be admittedly exploratory, avoiding the use of the word 'significant' and simplifying the analysis. Simple monthly averages and testing for the months having the same mean could replace the wavelet analysis. A second option would be to focus on a specific research hypothesis, explain carefully how the model estimates relate to this research hypothesis, and adjust the p-values for multiple testing.

---

## [Author Response]

Essential revisions:1) By choosing only 4 disease groups the authors seem to have missed a key opportunity to investigate seasonal patterns in finer subgroups despite possessing a large sample size (n = 77,771,264), Furthermore, the rationale for choosing the subgroups presented is unclear. For example, the authors chose to group cardiorespiratory diseases into one large outcome when cardiovascular and respiratory outcomes have differing mechanisms in their association with heat or cold exposure. Within respiratory diseases itself, it makes sense to separate acute and chronic causes because the mechanisms vary so starkly and are related to seasonal patterns, i.e. the incidence of pneumonia and acute respiratory outcomes are likely heightened in the winter and this observation may differ to how/why the incidence of chronic respiratory deaths vary seasonally. Furthermore, the mechanisms for how heat is associated with elevated respiratory deaths is understudied and knowing more about seasonal patterns here would be a useful addition to the literature.I believe it's important to investigate seasonal patterns of deaths from infectious disease, maternal and neonatal causes, endocrine disorders, genitourinary conditions and neuropsychiatric conditions (which are all associated with temperature) to add valuable insight into understudied but important disease groups, but these have been excluded from the present analysis. Considering the tragic opioid epidemic in USA, it would be a valuable contribution to the literature to understand whether seasonal patterns (at least in more recent years) are observed with deaths related to substance use disorders.

We had selected the four cause groups to go beyond all-cause mortality and still have parsimonious presentation and sufficient number of events by sex, age group, year, month and geography. In the specific case of cardiorespiratory diseases, they are commonly analysed together when studying the effects of temperature and air pollution (Basu, 2009; Basu, Dominici, and Samet, 2005; Basu and Samet, 2002; Bennett et al., 2014; Braga, Zanobetti, and Schwartz, 2002; Curriero et al., 2002; Dockery et al., 1993; Gasparrini et al., 2015; Hoek et al., 2013; Pope et al., 2002), possibly because of their shared aetiology (e.g., death from cor pulmonale arising from COPD or death from pneumonia in patients with acute vascular events) and because assignment of cause of death may use them interchangeably. We now present a third layer of disaggregation following the above suggestions (figure supplements for Figures 6-13; Table 1; Materials and methods).

2) The authors pointed out in the Introduction that global warming may impact on excess cold weather death rates. Presumably a major part of the rationale for testing the differences in regions over time was to help inform our understanding of what the impact may be. However no conclusions were reached with regard to the implications of the findings. Clearly any such conclusions would need to be appropriately and heavily caveated but as this is a major reason for the analysis. Please expand the Discussion to include these points.

To quantitatively analyse the potential implications of global climate change, one would need to formally incorporate temperature in the analysis which would involve a distinct analytical framework and presentation. As correctly pointed out in this comment, the current analysis can nonetheless provide qualitative insights into such effects, which we have now discussed with appropriate caveats (Discussion, second, third and fourth paragraphs).

3) Methodology:A) Wavelet power spectra are not always easy to interpret, and the uncertainty in estimated wavelet coefficients is difficult to quantify. The wavelet analysis makes for an interesting exploratory tool, showing that for the most part cycles have a duration of 12 months and are reasonably stable over time. It does not seem possible to draw firm conclusions about the research hypothesis using wavelets, however. All cause male mortality for 15-24 year olds appears to be less cyclical in recent years, although how to quantify this effect and assign a statistical significance to it is not apparent from the power spectrum shown.

Wavelet analysis is a formal analytical framework that can provide quantitative conclusions (Hubbard, 1996; Torrence and Compo, 1998), and has been applied in different areas of health and the environment, for example measles epidemics and El Niño oscillations, to quantitatively characterise their seasonality including.(Grenfell, Bjørnstad, and Kappey, 2001; Moyet al., 2002) A key advantage of wavelet analysis is that it does not assume a stationary time series, and hence can easily capture a decline/increase in or appearance/disappearance of seasonality. The uncertainty of wavelet spectra are also well-characterised (Hubbard, 1996). We follow wavelet analysis with a centre of gravity analysis to identify the months of maximum and minimum mortality (Figure 12 and respective figure supplements; subsection “Statistical methods"), and with a quantitative analysis of (percentage) difference in death rates from 1980 to 2016 (Figure 13 and respective figure supplements; subsection “Statistical methods").

We note that even in standard statistical comparisons, p-values and statistical significance are not apparent from figures but are a part of the analysis. We have added p-values to each of the wavelet analyses in Figures 2-11 and the respective figure supplements (subsection “Statistical methods").

The second analysis is more appropriate, although the details of this analysis are sparse. It appears that the model used is something like the following, where Yit is the count of deaths in age group i at time t.--------------------- -------------------------------------Yit∼ Poisson(Nit ⋅ λit)log(λit)= μi + βit + Mit ⋅ [α1i ⋅ cos(2πt/12)+α2i ⋅ sin(2πt/12)+α3i ⋅ cos(2πt/6)+α4i ⋅ sin(2πt/6)]Mit= ρi + γit--------------------- -------------------------------------A model of this could answer the following questions, with p-values produced from a likelihood ratio test.- Is γi negative? If so the seasonal effect is becoming less severe for age group i.- Does it look like all age groups have the same trend, with γi = γ₀ for all i?- Do all age groups have the same cycle, with αpi = αp0?- Does γ vary by region (negative in cold-climate regions?)- Do the cycles, given by the α, vary by region or is a nation-wide cycle sufficient?The above model is easier to interpret than the wavelet analysis. The quantity 1 − exp(120γ) is the change per decade in the seasonality effect, which could be reported with a 95% confidence interval.

Our current methods successfully answer the questions that the reviewer has posed. We have clarified our methods, especially those for Figures 12 to 17 and their respective figure supplements where applicable (referred to as the ‘second analysis’ above) in the revised manuscript and summarise the approach and its findings below.

- Is γi negative? If so the seasonal effect is becoming less severe for age group i.- Does it look like all age groups have the same trend, with γi = γ₀ for all i?

We calculated change in percent difference between 1980 and 2016 by fitting a linear regression to the time series of seasonal differences in mortality within a year. We have reported the percentage point change in seasonality, with p-values, from 1980 to 2016 by age, sex, and cause of death (Figure 13 and respective figure supplements).

- Do all age groups have the same cycle, with αpi = αp0?

We calculated the timing of maximum and minimum mortality using centre of gravity analysis and circular statistics. We have reported the maximum and minimum values by age, sex, and cause of death (Figure 12 and respective figure supplements).

- Does γ vary by region (negative in cold-climate regions)?

We report percent difference in death rates between maximum and minimum months by region (Figure 18 and subsection “Statistical methods"). We have also calculated change in percent difference between 1980 and 2016 by region, shown in Author response image 1 for all-cause mortality, though not included in the main paper which is already dense with inclusion of a large number of results at the national level.

**Author response image 1. respfig1:** Percent difference in death rates between the maximum and minimum mortality months for all-cause mortality in 2016 versus 1980 by sex, age group and region.

- Do the cycles, given by the α, vary by region or is a nation-wide cycle sufficient?

We calculated the timing of maximum and minimum mortality using centre of gravity analysis by region as well as nationally. We have reported the maximum and minimum values by age and sex for all-cause death (Figures 14-17 and subsection “Statistical methods").

Therefore, we have successfully answered all the questions posed by the reviewers with our current methods. The wavelet analysis allows greater flexibility when compared to the above model suggested because it does not require assuming only 6- and 12-month periodicity.

We have nonetheless implemented the above approach. We note that some the parameters of this model (e.g., the ρ, γ, and α terms) cannot be interpreted on their own, because the model includes involve interaction terms, for example (γ x α1i). In addition, the interpretation of γ is also dependent on the sign of α, as, for example, a negative value of α with a negative value of γ indicates the percentage difference between maximum and minimum mortality is increasing, whereas a positive value of α with a negative value of γ indicates that the percentage difference between maximum and minimum mortality is decreasing. For these reasons, interpretable results require predicting the outcomes in specific years and not simply the model parameters. We have done so for one of the outcomes presented in the paper, namely percent difference in death rates between maximum and minimum mortality months, with results presented in the table below. Conclusions were similar to those from the existing method.

We believe that our current approach has the dual advantage over the proposed method of using a more general and flexible framework and being more easily understandable. We welcome guidance on whether adding this additional method would improve the paper.

**Sex****Age (years)****Percent difference 1980****Percent difference 2016**Male0-411.96.9Female0-410.66.9Male5-1456.219.4Female5-1426.320.9Male15-2439.110.8Female15-2414.92.3Male25-3417.37Female25-346.94.3Male35-444.94Female35-446.57.1Male45-5410.97.9Female45-5411.411.2Male55-6413.711.7Female55-6413.113.3Male65-7417.614.5Female65-7416.615.4Male75-8424.320.2Female75-8424.621.2Male85+35.225.5Female85+34.227.6

**Author response table 1.** Percent difference in death rates between the maximum and minimum mortality months for all-cause mortality in 2016 versus 1980 by sex and age group using the alternative method.

B) The approach is different to a number of those referenced (particularly the standard measure of excess winter deaths). It would be helpful to have some comment on this, particularly the generalisability to other studies. The focus has been on peak months in this analysis, but the standard assessment of excess winter deaths is to compare December to March months to the rest of the year.

We have avoided the term Excess Winter Deaths (EWDs) because the latter assumes higher deaths in winter than in the summer, and requires a priori decisions about which months to include in “winter” and “summer”. Our empirical results show that peak mortality not only varies from age group to age group or cause by cause, but also can in the extreme take place in summer, e.g., for young adult males. We have added a brief overview of EWDs, and why an empirical approach like ours is preferred to its strong assumptions (P. 19).

C) Appropriate rationale was given to the regional splits used in the analysis. However no description was given of the characteristics of these regional areas other than subsequent information on temperatures. As pointed out there could be a number of factors that are important. I would expect some reference to their differing characteristics.

A fair comment and addressed in the revised manuscript by providing an overview of the characteristics of the regions (subsection “Data”, second paragraph and Table 2).

4) Interpretation:A) No table is presented with descriptive statistics, i.e. on the sample size of cases that fall into each disease category.

We have added a table of descriptive statistics (Table 1).

B) I may be misreading the paper (in which case it would be helpful to clarify so others do not make the same mistake), however the analysis in Figure 6 compares the difference in temperature experienced in those regions between the warmest and coldest months versus the% seasonal difference in death. In the Discussion this in contrasted with findings from Europe where countries with a more temperate winter have, paradoxically, higher rates of excess winter mortality. However this is not an appropriate direct comparison, regional extremes in variation were not looked at when comparisons between countries are made. This does not invalidate the comparison but it would be helpful to explicitly summarise if there are any differences between regions and examine what those are.

A fair comment and addressed in the revised manuscript in the Discussion. Specifically, the European papers that we had cited (Fowler et al., 2015; Healy, 2003; McKee, 1989) had based their comparison on annual mean temperature whereas ours was based on the temperature range. We have clarified this distinction in the revised Discussion, noting that the two temperature metrics are correlated (see Author response image 2).

**Author response image 2. respfig2:** The relationship between annual mean temperature (used in European papers) and temperature range between maximum and minimum mortality months (used in our paper).

C) One option for this paper would be to be admittedly exploratory, avoiding the use of the word 'significant' and simplifying the analysis. Simple monthly averages and testing for the months having the same mean could replace the wavelet analysis. A second option would be to focus on a specific research hypothesis, explain carefully how the model estimates relate to this research hypothesis, and adjust the p-values for multiple testing.

As above, wavelet analysis, the subsequent steps of centre-of-gravity analysis to identify peak and minimum mortality months, and estimation of changes in seasonal mortality range are all formal analyses.

We now quote p-values in results, as opposed to setting an explicit threshold for significance (as derived by comparing against a threshold on the p-value), hence removing the need for correction for multiple testing.